# Dual-Calibration Multi-View Clustering via Compact Anchor Learning

**Huibing Wang**[1]  **Yuemeng Huang**[1]  **Yawei Chen**[2]  **Jiaxin Yang**[1]  **Qian Liu**[1]  **Jinjia Peng**[*3]  **Zetian Mi**[1]
**Ximing Li**[2]

## Abstract

The anchor-based multi-view clustering methods have received extensive attention due to their efficiency and scalability in large-scale data scenarios. Existing anchor-based methods still face challenges in learning compact and semantically discriminative anchors. Current mainstream approaches typically rely on random sampling strategies or orthogonal constraints for anchor selection and learning. However, they often optimize anchor learning and cluster assignment in a relatively separate manner, leaving the clustering semantics in the sample space insufficiently exploited for calibrating the anchor space. As a result, the learned anchors may suffer from redundant coverage and ambiguous cluster boundaries. Unlike existing anchor-based multi-view clustering methods, this paper proposes a Dual-Calibration Multi-view Clustering via Compact Anchor Learning (DCMC), which effectively improves anchor quality through a dual-space alignment mechanism. Specifically, DCMC initializes view-specific anchors to capture the underlying data distribution, and then enforces bidirectional consistency between the anchor space and the clustering space to jointly optimize both the sample-to-anchor assignments and the cluster assignments. The alternating optimization process effectively enhances cross-view semantic consistency while preserving the discriminative characteristics of each view. Experimental results demonstrate that DCMC outperforms state-of-the-art methods across multiple benchmark tests, confirming its effectiveness and reliability.

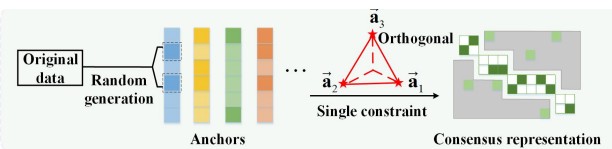

**(a) Current anchor learning method**

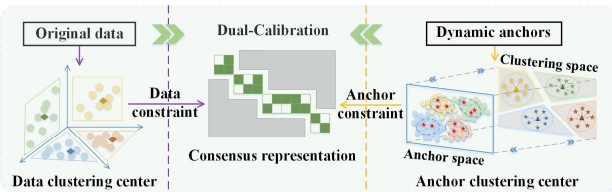

**(b) Dual-Calibration anchor learning**

*Figure 1.* Comparison of anchor learning strategies between existing methods and our method. (a) Existing methods often generate anchors from the original data and learn the consensus representation with a single anchor constraint. (b) Our method establishes a dual-calibration mechanism between anchor representation learning and sample clustering semantics, resulting in a more compact and discriminative clustering structure.

## 1. Introduction

With the increasing diversity of real-world data, multi-view learning has become crucial for integrating information from multiple perspectives (Sun et al., 2025; Yuan et al., 2025; Wang et al., 2026b). It effectively combines data correlations across different views to discover consistent and complementary features (Jiang et al., 2026). This approach reveals underlying semantic patterns while overcoming the limitations of single-view representations (Wang et al., 2025; Sun et al., 2024; Chen et al., 2024; 2025)

Based on these principles, multi-view learning has been widely applied to diverse learning tasks, among which multi-view clustering serves as a canonical unsupervised learning approach (Chen et al., 2023a; Cui et al., 2023; Wang et al., 2026a). In current multi-view clustering research, subspace-based methods have become prevalent owing to their superior ability to uncover latent relationships across multiple views (Lan et al., 2021; Zhang et al., 2020). However, many subspace methods suffer from high computational complexity, making them difficult to scale to large-scale datasets.

---

[1]School of Information Science and Technology, Dalian Maritime University, Dalian, China [2]School of Computer Science and Technology, Jilin University, Changchun, China [3]School of Cyber Security and Computer, Hebei University, Baoding, China. Correspondence to: Jinjia Peng <pengjinjia@hbu.edu.cn>.

*Proceedings of the 43rd International Conference on Machine Learning*, Seoul, South Korea. PMLR 306, 2026. Copyright 2026 by the author(s).

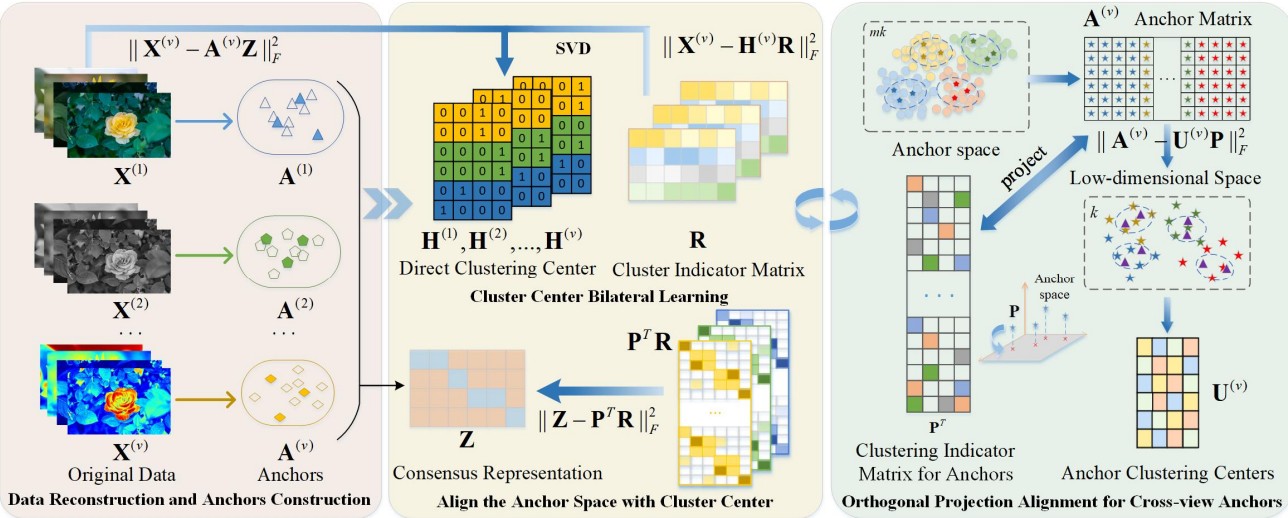

*Figure 2.* The DCMC method establishes a multi-view anchor construction framework with a dual-calibration mechanism that synchronously optimizes both data and anchor cluster centers in a consensus space. By enforcing orthogonal constraints, the method alleviates cross-view discrepancies and promotes compact semantic alignment. The core innovation lies in the joint learning of data reconstruction, anchor clustering, and consensus representation, ultimately attaining globally consistent clustering.

(Li et al., 2024) factorizes an anchor graph with orthogonal non-negative constraints to directly obtain cluster labels without post-processing, significantly reducing computational cost. (Yang et al., 2022b) designs a lightweight anchor graph with maximum correntropy criterion to enhance robustness against complex noise, while (Yang et al., 2022a) independently constructs anchor graphs for each view and incorporates orthogonal constraints to improve clustering efficiency. Although these methods have achieved promising clustering performance, the above methods usually generate or select anchors from the original data and then learn the consensus representation under fixed anchor constraints. They treat anchor construction and clustering structure learning in a relatively decoupled manner, making the learned anchors insufficiently aligned with clustering semantics. As illustrated in Figure 1(a), this may result in redundant anchor coverage and ambiguous cluster boundaries.

To address the above challenges, we propose Dual-Calibration Multi-view Clustering via Compact Anchor Learning (DCMC), as illustrated in Figure 2. DCMC learns compact and semantically discriminative anchors by jointly modeling anchor representation learning and data clustering semantics. Specifically, a compact cross-view anchor learning module is designed: by enforcing dual-space alignment constraints, semantic relationships between different view anchors are established, and the distributional consistency between anchors of different views is adjusted. On this basis, an iterative anchor optimization mechanism is introduced to guide the updating of anchor representations, ensuring that the anchors preserve view specificity while satisfying cross-view consistency, which significantly improves their

discriminative capability. Eventually, the optimized anchors from all views are integrated to construct a unified bipartite graph, upon which k-means is applied to obtain the final clustering results. The proposed objective promotes inter-class separability and intra-class compactness in the anchor space, leading to clearer cluster boundaries. The main contributions of the methodology can be summarized as follows:

- A novel anchor alignment strategy based on the reverse graph is proposed, which resolves the view deviation problem caused by the current independent generation of anchors, improving the cross-view consistency and discriminability of anchors.

- We design a dual-calibration framework to jointly calibrate anchor representations and sample clustering assignments in both the anchor space and the data space.

- Experimental results on benchmark datasets demonstrate the superior performance of our dual-calibration module over state-of-the-art methods.

## 2. Related Work

This section summarizes the key mathematical notations (Table 1) and reviews two existing related studies: multi-view subspace clustering and anchor graph-based multi-view clustering.

| Notations | Descriptions |
|-----------|--------------|
| $n, k, l$ | Number of samples, clusters and views |
| $m$ | Number of anchors per cluster |
| $d_v$ | Feature dimension of the $v$-th view |
| $\mathbf{X}^{(v)} \in \mathbb{R}^{d_v \times n}$ | Data matrix for the $v$-th view |
| $\mathbf{A}^{(v)} \in \mathbb{R}^{d_v \times mk}$ | Anchor matrix for the $v$-th view |
| $\mathbf{Z} \in \mathbb{R}^{mk \times n}$ | Consensus subspace representation |
| $\mathbf{U}^{(v)} \in \mathbb{R}^{d_v \times k}$ | Anchor clustering centers for the $v$-th view |
| $\mathbf{H}^{(v)} \in \mathbb{R}^{d_v \times k}$ | Clustering center of data for $v$-th view |
| $\mathbf{P} \in \mathbb{R}^{k \times mk}$ | Clustering assignment matrix for anchors |
| $\mathbf{R} \in \mathbb{R}^{k \times n}$ | Clustering assignment matrix for data |
| $\mathbf{I}_k \in \mathbb{R}^{k \times k}$ | Identity matrix |
| $\mathbf{I}_{mk} \in \mathbb{R}^{mk \times mk}$ | Identity matrix |

*Table 1.* Notations and Descriptions of The Main Formulas

## 2.1. Multi-view Subspace Clustering

Recently, multi-view subspace clustering (MVSC) has attracted growing interest for its ability to clearly represent data structures through low-rank representations, effectively capturing intrinsic structures while boosting efficiency. A typical MVSC objective function follows:

$$\min_{\mathbf{C}^{(v)}, \mathbf{C}} \sum_{v=1}^{l} \underbrace{||\mathbf{X}^{(v)} - \mathbf{X}^{(v)}\mathbf{C}^{(v)}||_F^2}_{\text{Graph Construction}} + \underbrace{\Phi(\mathbf{C}^{(v)}, \mathbf{C})}_{\text{Fusion}} \quad (1)$$

$$\text{s.t. } diag(\mathbf{C}^{(v)}) = 0, diag(\mathbf{C}) = 0, \mathbf{C}_i^T \mathbf{1} = 1.$$

where the first term denotes the subspace representation learning term of the $v$-th view and $\Phi$ denotes the uniformly consistent regularization term that ensures a uniform structure across views.

Based on the above framework, many papers have proposed improved multi-view subspace clustering methods (Chen et al., 2022b; Wen et al., 2023; Wu et al., 2019). Specifically, (Luo et al., 2018) proposed CSMSC, a subspace method that separates common and complementary information using shared and view-specific representations to enhance robustness. Subsequent work further improved performance by constructing an adaptive consensus graph filter for cross-view feature smoothing and representation regularization (Wei & Song, 2024) and by introducing a Laplacian rank constraint to enforce k-connectivity of the consistent anchor graph (Liu et al., 2024). Furthermore, some methods stack the representation matrices into a tensor and utilize tensor decomposition to capture higher-order correlations among views.

Although the above methods improve the performance of clustering, notably, most of them are constrained by the time complexity of $\mathcal{O}(n^3)$. This is problematic when the data size is large, as it demands high computational resources.

## 2.2. Anchor graph-based multi-view clustering

In recent years, anchor graph-based multi-view clustering has gained significant popularity due to its computational efficiency (Wang et al., 2023). The core idea involves selecting representative anchors to replace traditional sample-sample relationships with anchor-sample relationships, reducing complexity from $\mathcal{O}(n^2)$ to $\mathcal{O}(mn)$ while maintaining clustering performance, making it suitable for large-scale datasets.

Recent research on anchor graph-based MVC methods has made new progress. (Qin et al., 2023) combines row space and column space of data to learn information-rich anchors. (Li et al., 2023) introduces tensor Schatten p-norm regularization to further mine complementary information between views. (Mi et al., 2024) proposed a fast clustering method to dynamically learn anchors and anchor graphs in the embedding space instead of the original space, effectively eliminating the effect of noise.

While these methods demonstrate measurable improvements over baseline approaches, several limitations persist that warrant further investigation. Different from existing methods that mainly improve the sample-anchor graph or impose consistency on view-specific graphs, DCMC explicitly couples anchor learning with data clustering assignments. This design allows the anchors to be calibrated not only by reconstruction errors but also by clustering semantics, thereby reducing redundant anchor coverage and ambiguous cluster boundaries.

## 3. The Proposed Method

This section elaborates on the DCMC method, including its mathematical formulation, optimization algorithm, and complexity analysis.

### 3.1. Proposed Formula

In multi-view learning tasks, data from different views typically possess high-dimensional heterogeneous features. Based on the theoretical foundation of the manifold assumption, this study hypothesizes that all views share a unified consensus low-dimensional subspace. Consequently, the mathematical formulation for anchor learning and anchor graph construction can be derived as follows.

$$\min_{\mathbf{A}^{(v)}, \mathbf{Z}} \sum_{v=1}^{l} \left\| \mathbf{X}^{(v)} - \mathbf{A}^{(v)}\mathbf{Z} \right\|_F^2. \quad (2)$$

where $\mathbf{A}^{(v)}$ is the view-specific anchors of $v$-th view and $\mathbf{Z}$ denotes the consensus anchor representation.

***Compact Cross-view Anchor Learning:*** Current approaches often face a fundamental challenge – achieving an optimal balance between enforcing inter-view consistency

and preserving view-specific characteristics. To address this critical issue, this study proposes an orthogonal projection decomposition framework, which aims to simultaneously satisfy the dual requirements of cross-view anchor alignment and view-specific feature preservation by mapping heterogeneous view-specific anchor spaces into a unified clustering subspace. Based on this theoretical foundation, we formally express the core optimization objective as:

$$\min_{\mathbf{P}, \mathbf{U}^{(v)}} \sum_{v=1}^{l} \left\| \mathbf{A}^{(v)} - \mathbf{U}^{(v)}\mathbf{P} \right\|_F^2 \quad \text{s.t. } \mathbf{U}^{(v)T}\mathbf{U}^{(v)} = \mathbf{I}.$$
(3)

where $\mathbf{A}^{(v)}$ is factorized into view-specific basis $\mathbf{U}^{(v)}$ and clustering indicator matrix for anchors $\mathbf{P}$. An orthogonal constraint is imposed on $\mathbf{U}^{(v)}$ to ensure the independence of projection directions. The intuitive meaning of this decomposition is that $\mathbf{U}^{(v)}$ encodes the view-specific latent projection directions of the $v$-th view, while $\mathbf{P}$ denotes the anchor-cluster assignment matrix, which maps anchors to latent cluster centers, thereby achieving semantic alignment and structural unification among multiple views while preserving their distinctive characteristics. The decomposition mechanism works in concert with the data reconstruction objective. The dual constraints imposed on the anchors not only ensure more accurate data representation but also enable different views to share a consistent clustering structure while preserving their view-specific cluster center characteristics. This approach effectively maintains multi-view consistency while retaining the distinctive features of each individual view.

*Dual-Calibration Latent Representation Learning:* Based on the orthogonal projection decomposition framework established Eq.(3), this paper further proposes a bidirectional calibration module. It aims to enhance the discriminability and stability of latent representations through bidirectional consistency constraints between the anchor space and the original data space, thereby achieving synergistic improvement between representation learning and clustering optimization. On the one hand, the clustering structure of each view is learned in the original data space; on the other hand, sample representations are calibrated in the anchor space, forming a bidirectional optimization process. Specifically, the bidirectional calibration module can be expressed as follows:

$$\min_{\mathbf{R}, \mathbf{H}^{(v)}, \mathbf{Z}} \sum_{v=1}^{l} \left\| \mathbf{X}^{(v)} - \mathbf{H}^{(v)}\mathbf{R} \right\|_F^2 + \left\| \mathbf{Z} - \mathbf{P}^T\mathbf{R} \right\|_F^2$$
$$\text{s.t. } \mathbf{H}^{(v)T}\mathbf{H}^{(v)} = \mathbf{I}, \mathbf{R}^T\mathbf{1} = \mathbf{1}, \mathbf{R} \geq 0.$$
(4)

where $\mathbf{H}^{(v)}$ represents the cluster centers of the original data in the $v$-th view, and the first term is designed to learn the cluster centers $\mathbf{H}^{(v)}$ and the cluster indicator matrix $\mathbf{R}$ for each view from the original data. The second term enforces

the association between the anchor representation $\mathbf{Z}$ and the data clustering result $\mathbf{R}$ through $\mathbf{P}^T$, where $\mathbf{R}$ refines the anchor representation $\mathbf{Z}$, thereby influencing the anchor matrix $\mathbf{A}^{(v)}$. Consequently, a reverse reconstruction from data to anchors is achieved.

Combining the above formulations, we obtain the overall objective function of our proposed method as:

$$\min_{\Phi} \sum_{v=1}^{l} \left\| \mathbf{X}^{(v)} - \mathbf{A}^{(v)}\mathbf{Z} \right\|_F^2 + \lambda_1 \sum_{v=1}^{l} \left\| \mathbf{A}^{(v)} - \mathbf{U}^{(v)}\mathbf{P} \right\|_F^2$$
$$+ \sum_{v=1}^{l} \left\| \mathbf{X}^{(v)} - \mathbf{H}^{(v)}\mathbf{R} \right\|_F^2 + \lambda_2 \left\| \mathbf{Z} - \mathbf{P}^T\mathbf{R} \right\|_F^2$$
$$\text{s.t. } \mathbf{U}^{(v)T}\mathbf{U}^{(v)} = \mathbf{I}, \mathbf{H}^{(v)T}\mathbf{H}^{(v)} = \mathbf{I}, \mathbf{R}^T\mathbf{1} = \mathbf{1}, \mathbf{R} \geq 0.$$
(5)

where $\lambda_1$ and $\lambda_2$ are balancing parameters. For notational simplicity, we denote all optimization variables as $\Phi = \{\{\mathbf{A}^{(v)}, \mathbf{U}^{(v)}, \mathbf{H}^{(v)}\}_{v=1}^{V}, \mathbf{Z}, \mathbf{P}, \mathbf{R}\}$. The first term learns anchors and the anchor graph through data reconstruction to obtain a representative latent anchor structure; the second term enforces alignment between the anchor representation and the clustering results of anchors; the third term directly optimizes the cluster centers and sample assignments for each view in the original data space to improve clustering discriminability; the fourth term enhances the compactness and cross-view stability of the representation by aligning the anchor representation with clustering semantics and enforcing consistency among anchors across different views.

### 3.2. Optimization

Since the objective function in Eq.(5) exhibits non-convexity when all variables are jointly optimized, directly obtaining the global optimal solution becomes computationally challenging. To address this, we employ the alternating minimization to decompose the optimization problem.

**1) $\mathbf{H}^{(v)}$ Sub-Step:** With all other variables held fixed, the subproblem with respect to $\mathbf{H}^{(v)}$ is formulated as the following optimization problem:

$$\min_{\mathbf{H}^{(v)}} \left\| \mathbf{X}^{(v)} - \mathbf{H}^{(v)}\mathbf{R} \right\|_F^2, \quad \text{s.t. } \mathbf{H}^{(v)T}\mathbf{H}^{(v)} = \mathbf{I}.$$
(6)

To derive a tractable equivalent formulation, we exploit the relationship between the Frobenius norm and the trace to expand the objective function. Under the orthogonality constraint, minimizing the reconstruction error is equivalent to the following form:

$$\max_{\mathbf{H}^{(v)}} \text{Tr}\left( \mathbf{H}^{(v)T}\mathbf{X}^{(v)}\mathbf{R}^T \right), \quad \text{s.t. } \mathbf{H}^{(v)T}\mathbf{H}^{(v)} = \mathbf{I}.$$
(7)

Due to the orthogonality constraint, the optimal solution $\mathbf{H}^{(v)}$ can be obtained through singular value decomposition

to maximize the trace. Let the singular value decomposition of $\mathbf{X}^{(v)}\mathbf{R}^T$ be $\mathbf{X}^{(v)}\mathbf{R}^T = \mathbf{M}\mathbf{\Sigma}\mathbf{N}^T$. Then, the optimal solution is given by $\mathbf{H}^{(v)} = \mathbf{M}\mathbf{N}^T$.

**2) $\mathbf{A}^{(v)}$ Sub-Step:** With all other variables held fixed, the subproblem with respect to $\mathbf{A}^{(v)}$ is formulated as the following optimization problem:

$$\min_{\mathbf{A}^{(v)}} \left\| \mathbf{X}^{(v)} - \mathbf{A}^{(v)}\mathbf{Z} \right\|_F^2 + \lambda_1 \left\| \mathbf{A}^{(v)} - \mathbf{U}^{(v)}\mathbf{P} \right\|_F^2 \tag{8}$$

By taking the derivative of $\mathbf{A}^{(v)}$ and setting it to zero yields, we can obtain:

$$\mathbf{A}^{(v)}\mathbf{Z}\mathbf{Z}^T + \lambda_1\mathbf{A}^{(v)} - \lambda_1\mathbf{U}^{(v)}\mathbf{P} - \mathbf{X}^{(v)}\mathbf{Z}^T = \mathbf{0} \tag{9}$$

After simplification, the analytical solution can be easily obtained:

$$\mathbf{A}^{(v)} = \left( \lambda_1\mathbf{U}^{(v)}\mathbf{P} + \mathbf{X}^{(v)}\mathbf{Z}^T \right) \left( \mathbf{Z}\mathbf{Z}^T + \lambda_1\mathbf{I}_{mk} \right)^{-1} \tag{10}$$

**3) $\mathbf{U}^{(v)}$ Sub-Step:** The optimization problem with respect to $\mathbf{U}^{(v)}$ can be formulated as:

$$\min_{\mathbf{U}^{(v)}} \left\| \mathbf{A}^{(v)} - \mathbf{U}^{(v)}\mathbf{P} \right\|_F^2, \quad \text{s.t. } \mathbf{U}^{(v)T}\mathbf{U}^{(v)} = \mathbf{I}. \tag{11}$$

By expanding the Frobenius norm and using the properties of the trace, we obtain the following.

$$\max_{\mathbf{U}^{(v)}} \mathrm{Tr}\left( \mathbf{U}^{(v)T}\mathbf{A}^{(v)}\mathbf{P}^T \right), \quad \text{s.t. } \mathbf{U}^{(v)T}\mathbf{U}^{(v)} = \mathbf{I}. \tag{12}$$

We can obtain the optimal solution $\mathbf{U}^{(v)}$ that maximizes the trace, i.e., $\mathbf{A}^{(v)}\mathbf{P}^T = \mathbf{W}\mathbf{\Sigma}\mathbf{V}^T$. The optimal solution is $\mathbf{U}^{(v)} = \mathbf{W}\mathbf{V}^T$.

**4) $\mathbf{P}$ Sub-Step:** The optimization problem with respect to $\mathbf{P}$ can be formulated as follows:

$$\min_{\mathbf{P}} \sum_{v=1}^{l} \left( \lambda_1 \left\| \mathbf{A}^{(v)} - \mathbf{U}^{(v)}\mathbf{P} \right\|_F^2 \right) + \lambda_2 \left\| \mathbf{Z} - \mathbf{P}^T\mathbf{R} \right\|_F^2 \tag{13}$$

By applying the matrix differentiation rules to P and setting the derivative to zero, we can obtain the following after simplification:

$$\mathbf{P} = \left( \lambda_1 l\mathbf{I}_k + \lambda_2\mathbf{R}\mathbf{R}^T \right)^{-1} \left( \lambda_1 \sum_{v=1}^{l} \mathbf{U}^{(v)T}\mathbf{A}^{(v)} + \lambda_2\mathbf{R}\mathbf{Z}^T \right) \tag{14}$$

**5) $\mathbf{Z}$ Sub-Step:** The optimization problem with respect to $\mathbf{Z}$ can be formulated as follows:

$$\min_{\mathbf{Z}} \sum_{v=1}^{l} \left\| \mathbf{X}^{(v)} - \mathbf{A}^{(v)}\mathbf{Z} \right\|_F^2 + \lambda_2 \left\| \mathbf{Z} - \mathbf{P}^T\mathbf{R} \right\|_F^2. \tag{15}$$

By decomposing the reconstruction term into the sum of squared errors for all samples, the global optimization problem is transformed into independent constraint optimization

**Algorithm 1 DCMC algorithm**

**Input**: Multi-view data $\{\mathbf{X}^{(v)}\}_{v=1}^{l}$, cluster number $k$, $\lambda_1$, $\lambda_2$, maximum iterations MAX_ITER
**Output**: Perform k-means on $\mathbf{Z}$ to obtain the clusters.

1: Initialize: $\mathbf{A}^{(v)}$, $\mathbf{R}$, $\mathbf{P}$
2: **while** not converged **do**
3:     Update variable $\mathbf{Z}$ using Eq. (17);
4:     Update $\mathbf{H}^{(v)}$ by solving the orthogonal procrustes problem in Eq. (7) via SVD;
5:     Update variable $\mathbf{R}$ using Eq. (20);
6:     Update variable $\mathbf{A}^{(v)}$ using Eq. (10);
7:     Update $\mathbf{U}^{(v)}$ by solving the orthogonal procrustes problem in Eq. (12) via SVD;
8:     Update variable $\mathbf{P}$ using Eq. (14);
9: **end while**

for each individual sample. Meanwhile, the anchor regularization term remains unchanged in its original form.

$$\min_{\mathbf{Z}} \sum_{v=1}^{l} \sum_{i=1}^{n} \left\| x_i^{(v)} - \mathbf{A}^{(v)}z_i \right\|_2^2 + \lambda_2 \left\| \mathbf{Z} - \mathbf{P}^T\mathbf{R} \right\|_F^2 \tag{16}$$

By setting the derivative of Z to zero, we obtain the closed-form optimal solution.

$$\mathbf{Z} = \left( \sum_{v=1}^{l} \mathbf{A}^{(v)T}\mathbf{A}^{(v)} + \lambda_2\mathbf{I}_{mk} \right)^{-1} \left( \sum_{v=1}^{l} \mathbf{A}^{(v)T}\mathbf{X}^{(v)} + \lambda_2\mathbf{P}^T\mathbf{R} \right). \tag{17}$$

**6) $\mathbf{R}$ Sub-Step:** The optimization problem with respect to $\mathbf{R}$ can be formulated as follows:

$$\min_{\mathbf{R}} \sum_{v=1}^{l} \left\| \mathbf{X}^{(v)} - \mathbf{H}^{(v)}\mathbf{R} \right\|_F^2 + \lambda_2 \left\| \mathbf{Z} - \mathbf{P}^T\mathbf{R} \right\|_F^2, \tag{18}$$
$$\text{s.t. } \mathbf{R}^T\mathbf{1} = \mathbf{1}, \mathbf{R} \geq 0.$$

To proceed with the solution, we convert the Frobenius norm into the form of matrix trace, expand and rearrange the expression, and obtain the following equivalent formulation:

$$\min_{\mathbf{R}} \mathrm{Tr}\left( \mathbf{R}^T \left( \sum_{v=1}^{l} \mathbf{H}^{(v)T}\mathbf{H}^{(v)} + \lambda_2\mathbf{P}\mathbf{P}^T \right) \mathbf{R} - 2 \left( \sum_{v=1}^{l} \mathbf{H}^{(v)T}\mathbf{X}^{(v)} + \lambda_2\mathbf{P}\mathbf{Z} \right) \mathbf{R}^T \right). \tag{19}$$

We decompose the global matrix optimization into independent subproblems by reformulating the Frobenius-norm objective into element-wise quadratic programs. This reformulation decomposes the global optimization problem into

$n$ independent column-wise quadratic programming sub-problems, each corresponding to the soft cluster assignment of one sample.

$$
\min_{\mathbf{R}_{:,j}} \frac{1}{2} \mathbf{R}_{:,j}^T \left( \sum_{v=1}^{l} \mathbf{H}^{(v)T} \mathbf{H}^{(v)} + \lambda_2 \mathbf{P} \mathbf{P}^T \right) \mathbf{R}_{:,j}
$$
$$
- \left( \sum_{v=1}^{l} \mathbf{H}^{(v)T} \mathbf{X}_{:,j}^{(v)} + \lambda_2 \mathbf{P} \mathbf{Z}_{:,j} \right)^T \mathbf{R}_{:,j}. \tag{20}
$$

The above analysis demonstrates that the proposed method involves the iterative updating of six core variables. For each variable, we have developed corresponding optimization strategies. Algorithm 1 summarizes the detailed implementation steps of the entire optimization process.

### 3.3. Complexity Analysis

In this section, we conduct a comprehensive computational complexity analysis of the DCMC method, specifically examining its time complexity and space complexity.

**Time complexity:** The computational overhead is primarily determined by the cumulative time required for updating each variable. Specifically, the update of variable $\mathbf{Z}$ involves matrix inversion and multiplication operations, yielding an overall time complexity of $O(m^3 k^3 + l d_v m^2 k^2 + l d_v mkn + mk^2 n)$. For updating variable $\mathbf{A}^{(v)}$, the time complexity is $O(m^3 k^3 + m^2 k^2 n + d_v mkn)$. For the joint optimization of variables $\mathbf{P}$ and $\mathbf{R}$, the time complexity is $O(l d_v kn + k^2 n + mk^2 n)$. For updating variable $\mathbf{U}^{(v)}$ and $\mathbf{H}^{(v)}$, SVD decomposition is required, the time complexity is $O(l d_v k^2)$. Considering $m, k, d_v \ll n$, the overall time complexity can be approximated as $O(n)$, indicating linear time complexity.

**Memory complexity:** DCMC requires storing the following matrices: $\mathbf{X}^{(v)} \in \mathbb{R}^{d_v \times n}$, $\mathbf{A}^{(v)} \in \mathbb{R}^{d_v \times mk}$, $\mathbf{Z} \in \mathbb{R}^{mk \times n}$, $\mathbf{U}^{(v)}, \mathbf{H}^{(v)} \in \mathbb{R}^{d_v \times k}$, $\mathbf{P} \in \mathbb{R}^{k \times mk}$, $\mathbf{R} \in \mathbb{R}^{k \times n}$. The total space complexity can be derived as $\mathcal{O}(l \cdot d_v(n + mk) + mkn + k^2)$. When $n \gg mk$, it can be approximated as $\mathcal{O}(d_v n + kn)$, indicating linear space complexity.

## 4. Experiment

We evaluate the performance of our proposed DCMC method in terms of clustering results, parameter sensitivity, and computational efficiency, comparing it against nine state-of-the-art multi-view clustering approaches on five widely used benchmark datasets.

### 4.1. Experiment Settings

We compared the proposed DCMC method with 9 state-of-the-art multi-view clustering methods on 5 datasets, validating the performance of the proposed algorithm.

**Datasets:** (1) MSRC_v1(Chen et al., 2020) comprises 210

samples categorized into 7 distinct classes, with each sample containing 4 feature views. (2) **BBCSport**[1] consists of 544 sports news articles divided into 5 sport categories, where 2 feature views represent each article. (3) **Wiki**[2] is a Wikipedia article feature dataset containing 2,866 samples classified into 10 categories, with each sample having 2 feature views. (4) **MNIST**[3], a benchmark dataset for hand-written digit recognition, contains approximately 60,000 samples of digits 0-9, where each digit sample is represented as a 28×28 grayscale image. (5) **YouTubeFace_sel**[4] comprises 101,499 facial image samples covering 31 different individual identities, with each sample containing 5 feature views.

**Compared methods:** (1)LMVSC (Kang et al., 2020); (2)OMSC (Chen et al., 2022a); (3)FPMVS (Wang et al., 2021); (4)EOMSC (Liu et al., 2022); (5)AWMVC (Wan et al., 2023); (6)FDAGF (Zhang et al., 2023); (7)FSMSC (Chen et al., 2023b); (8)CAMVC (Zhang et al., 2024); (9)RCAGL (Liu et al., 2024).

To ensure a fair comparison of results, the hyperparameters of all methods were carefully tuned based on their original configurations. The model exhibits differential sensitivity across different datasets: on BBCSport, an increase in $\lambda_1$ leads to a notable performance decline, indicating that the view decomposition constraint plays a dominant role in this scenario, requiring careful control to avoid over-constraint; whereas on Wiki, the model shows stable and high-accuracy performance within moderate parameter ranges, with degradation occurring only under extreme parameter settings, demonstrating stronger parameter adaptability.

### 4.2. Experiment Results

Our method was compared with nine state-of-the-art approaches on five benchmark datasets, where LMVSC employs static anchor selection while the others adopt dynamic anchor strategies. Table 2 presents the comparative results using four evaluation metrics. Based on the experimental results, the following conclusions can be drawn: (1) Dynamic anchor-based methods, such as DCMC, CAMVC, RCAGL, and FSMSC, exhibit significantly better performance compared to LMVSC. This advantage stems from their adaptive adjustment capability and enhanced robustness against noise in dynamic anchor learning. (2) DCMC achieves optimal performance in most cases. Particularly in the MSRC_v1, BBCSport and Wiki dataset. (3) Most existing methods solely impose constraints on the anchor graph while neglecting the optimization of anchors, resulting in suboptimal anchor learning. In contrast, DCMC employs

---

[1]http://mlg.ucd.ie/datasets/bbc.html
[2]http://www.svcl.ucsd.edu/projects/crossmodal/
[3]https://yann.lecun.com/exdb/mnist/
[4]http://www.cs.tau.ac.il/ wolf/ytfaces/

| Dataset | Metrics | LMVSC | OMSC | FPMVS | EOMSC | AWMVC | FDAGF | FSMSC | CAMVC | RCAGL | Ours |
|---|---|---|---|---|---|---|---|---|---|---|---|
| MSRC_v1 | ACC | 0.3476 | 0.7048 | 0.6047 | 0.5905 | 0.7810 | 0.6429 | 0.7423 | 0.8418 | 0.8023 | **0.9333** |
| | NMI | 0.2461 | 0.6314 | 0.5557 | 0.4352 | 0.7160 | 0.6270 | 0.7026 | 0.8370 | 0.7968 | **0.8851** |
| | Purity | 0.4048 | 0.7381 | 0.6190 | 0.6048 | 0.7810 | 0.7571 | 0.7428 | 0.8466 | 0.7816 | **0.9333** |
| | F-score | 0.2480 | 0.6123 | 0.5029 | 0.4261 | 0.6852 | 0.5537 | 0.6467 | 0.8454 | 0.8000 | **0.8730** |
| BBCSport | ACC | 0.6544 | 0.4522 | 0.4209 | 0.4154 | 0.6397 | 0.4669 | 0.8698 | 0.8997 | 0.6959 | **0.9504** |
| | NMI | 0.4471 | 0.2129 | 0.1508 | 0.2367 | 0.4820 | 0.3749 | 0.8516 | **0.8951** | 0.5018 | 0.8579 |
| | Purity | 0.6636 | 0.5110 | 0.5183 | 0.5037 | 0.6890 | 0.9228 | 0.8780 | 0.9070 | 0.7018 | **0.9504** |
| | F-score | 0.4854 | 0.3556 | 0.3247 | 0.3468 | 0.5234 | 0.4978 | 0.8019 | 0.8683 | 0.6732 | **0.8999** |
| Wiki | ACC | 0.1884 | 0.3535 | 0.3140 | 0.5415 | 0.2170 | 0.5265 | 0.5872 | 0.5386 | 0.5078 | **0.6200** |
| | NMI | 0.0469 | 0.1991 | 0.1715 | 0.5290 | 0.0783 | 0.5149 | 0.5418 | 0.5339 | 0.4436 | **0.5696** |
| | Purity | 0.2083 | 0.3765 | 0.3367 | 0.6141 | 0.2547 | **0.6811** | 0.6015 | 0.5434 | 0.5178 | 0.6368 |
| | F-score | 0.1294 | 0.2243 | 0.2146 | 0.4832 | 0.1432 | 0.4325 | 0.5034 | 0.5543 | 0.4654 | **0.5560** |
| MNIST | ACC | 0.9896 | 0.8922 | 0.9884 | 0.9808 | 0.9885 | 0.9887 | **0.9954** | 0.9826 | 0.8863 | 0.9901 |
| | NMI | 0.9685 | 0.9336 | 0.9650 | 0.9475 | 0.9647 | 0.9658 | 0.9768 | **0.9787** | 0.9065 | 0.9701 |
| | Purity | 0.9896 | 0.8922 | 0.9768 | 0.9621 | 0.9761 | 0.9735 | 0.9735 | 0.9826 | 0.8998 | **0.9901** |
| | F-score | 0.9794 | 0.8923 | 0.9768 | 0.9621 | 0.9671 | 0.9735 | 0.9768 | 0.9787 | 0.8998 | **0.9803** |
| YouTubeFace_sel | ACC | 0.1479 | 0.1897 | **0.2414** | 0.1924 | 0.2367 | 0.2146 | 0.2398 | 0.1878 | 0.2098 | 0.2310 |
| | NMI | 0.1327 | 0.1225 | 0.1327 | 0.1154 | 0.2097 | 0.1966 | 0.0332 | 0.1957 | 0.2157 | **0.2160** |
| | Purity | 0.1216 | 0.1365 | **0.3279** | 0.1141 | 0.1547 | 0.0811 | 0.2689 | 0.2714 | 0.2895 | 0.3269 |
| | F-score | 0.0849 | 0.1243 | 0.1433 | 0.0832 | 0.1032 | 0.0732 | 0.1583 | 0.1413 | **0.1633** | 0.0952 |

*Table 2.* Clustering results on five datasets measured by four evaluation metrics. Optimal and suboptimal performances are denoted in **red** and blue, respectively.

joint optimization of both anchors and anchor graph, overcoming the uncontrollable anchor quality issue in existing methods, thereby yielding superior clustering results.

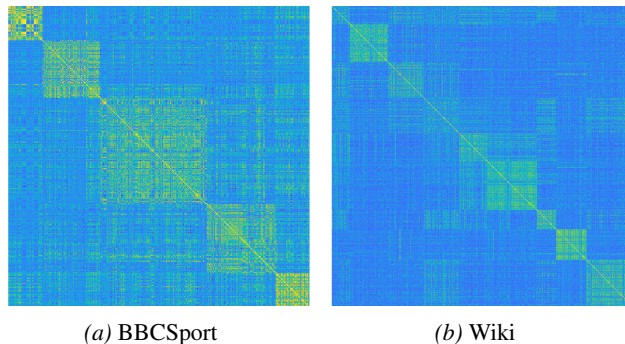

*(a)* BBCSport  *(b)* Wiki

*Figure 3.* The consensus graph generated by DCMC on both BBCSport and Wiki datasets.

### 4.3. Qualified Study and Parameter Analysis

To validate the superior performance of the consensus graph learned by DCMC, this study presents visualization results of the constructed consensus graph on BBCSport and Wiki datasets in Figure 3(a) and (b). Experimental evidence demonstrates that the consensus graph for both datasets exhibits pronounced block-diagonal structures. This compelling visualization provides confirmation that DCMC effectively learns consensus graphs with well-defined cluster-

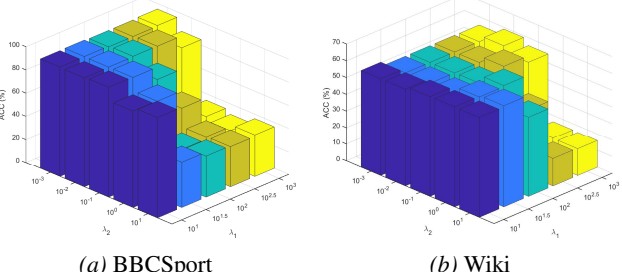

*(a)* BBCSport  *(b)* Wiki

*Figure 4.* ACC w.r.t $\lambda_1$ and $\lambda_2$ on BBCSport and Wiki datasets.

ing structures.

To further examine the robustness of the proposed model, a parameter sensitivity analysis is conducted by measuring the clustering accuracy (ACC) under various combinations of the trade-off parameters $\lambda_1 \in \{10^1, 10^{1.5}, 10^2, 10^{2.5}, 10^3\}$ and $\lambda_2 \in \{10^{-3}, 10^{-2}, 10^{-1}, 10^0, 10^1\}$, as illustrated in Figure 4. On the BBCSport dataset, the model demonstrates noticeable sensitivity to $\lambda_1$, with ACC varying considerably as the parameter changes, indicating a strong influence of the view-specific anchor decomposition term on clustering performance in this scenario. Conversely, on the Wiki dataset, the model exhibits high robustness, as ACC remains consistently stable throughout the tested parameter range, suggesting that the method maintains reliable clustering results under different regularization settings on this

dataset. These observations highlight the dataset-specific nature of parameter tuning and underscore the adaptability of the model structure to different data characteristics.

As can be seen from Figure 5, DCMC remains generally stable with respect to changes in the number of anchors on both datasets, with only small fluctuations. Among these, $m = 4k$ and $m = 5k$ usually yield slightly better results. Considering both performance and computational cost, the number of anchors in experiments can be set within the range $\{3k, 4k, 5k\}$.

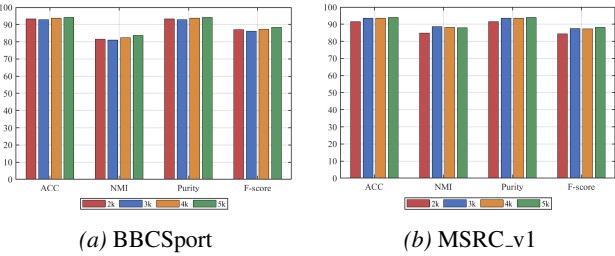

*(a)* BBCSport        *(b)* MSRC_v1

*Figure 5.* Sensitivity analysis of the number of anchors $m$ on BBCSport and MSRC_v1 datasets.

### 4.4. Convergence and Time Comparison

In this section, Figure 6 illustrates the trends of ACC and loss values with respect to the increasing number of iterations on the MSRC_v1 and BBCSport datasets. As shown in the figure, the objective value generally decreases and becomes stable within about 20 iterations.

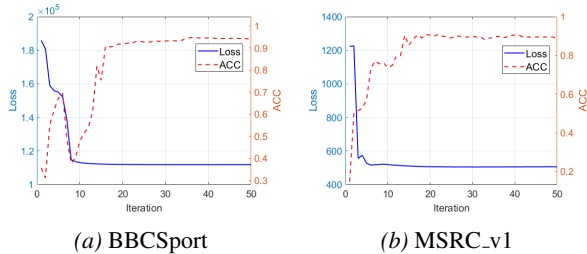

*(a)* BBCSport        *(b)* MSRC_v1

*Figure 6.* Convergence curves of DCMC on two datasets BBCSport and MSRC_v1.

To ensure comparability in computational efficiency across different methods, we conducted a comprehensive runtime analysis of all methods on five benchmark datasets. As shown in Figure 7, the results are presented on a logarithmic scale to clearly visualize the differences in computational time among the methods. Missing bars indicate that the corresponding method failed to complete due to memory overflow. This runtime analysis not only quantifies the actual computational overhead of each algorithm but also reflects their memory scalability, thereby providing empirical evidence for model selection in large-scale data scenarios.

The experiment reveals that while DCMC does not achieve

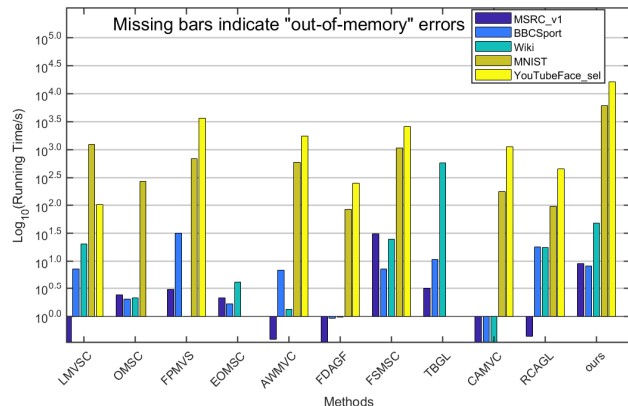

*Figure 7.* Comparative results of the execution time between our algorithm and nine benchmark methods across five datasets.

optimal runtime performance, the proposed dual-calibrated anchor selection strategy demonstrates marked advantages in enhancing anchor quality. Moreover, while maintaining controlled computational complexity, DCMC addresses limitations of existing methods, including redundant anchors and ambiguous clustering boundaries.

| Dataset | Method | ACC | NMI | Purity | F-score |
|---|---|---|---|---|---|
| MSRC_v1 | Ours-a | 0.1476 | 0.0548 | 0.1714 | 0.2430 |
| | Ours-b | 0.4238 | 0.2781 | 0.4238 | 0.2738 |
| | Ours | **0.9333** | **0.8851** | **0.9333** | **0.8730** |
| BBCSport | Ours-a | 0.6544 | 0.5803 | 0.6710 | 0.6360 |
| | Ours-b | 0.6544 | 0.5811 | 0.6728 | 0.6350 |
| | Ours | **0.9504** | **0.8579** | **0.9504** | **0.8999** |
| Wiki | Ours-a | 0.1581 | 0.0060 | 0.1595 | 0.1943 |
| | Ours-b | 0.1542 | 0.0191 | 0.1818 | 0.1148 |
| | Ours | **0.6200** | **0.5696** | **0.6368** | **0.5560** |
| MNIST | Ours-a | 0.6674 | 0.7609 | 0.6799 | 0.6249 |
| | Ours-b | 0.1124 | 0.0004 | 0.1125 | 0.1823 |
| | Ours | **0.9901** | **0.9701** | **0.9901** | **0.9803** |
| YouTubeFace_sel | Ours-a | **0.2663** | 0.0011 | 0.2665 | **0.1643** |
| | Ours-b | 0.2661 | 0.0010 | 0.2664 | 0.1642 |
| | Ours | 0.2310 | **0.2160** | **0.3269** | 0.0952 |

*Table 3.* Benchmarking results contrast the performance of Ours-a, Ours-b, and Ours on five established datasets.

### 4.5. Ablation Study

In this section, we perform ablation experiments on the DCMC method to verify the effectiveness of the dual-calibrated mechanism. We conduct experiments by respectively removing the anchor alignment module (Ours-a) and the consensus representation alignment module (Ours-b). Table 3 presents the comparison results of the variant methods and our method across five benchmark datasets.

As can be seen from the table, our method outperforms

variants on most datasets, demonstrating the dual-calibration module's role in ensuring cross-view anchor consistency while improving intra-class compactness and inter-class separation for enhanced robustness.

## 5. Conclusion

This paper proposes a Dual-Calibration Multi-view Clustering method via Compact Anchor Learning. Its core lies in an anchor alignment strategy that constrains both anchor graph and anchors, utilizing consensus representation to achieve synergistic optimization between the anchor space and the data space. Compared to existing approaches, our method achieves two key breakthroughs: first, it overcomes the limitations of single-path optimization through dual calibration across the data space and the anchor space; second, the designed bidirectional consistency constraints establish a dynamic correlation between anchor clustering and data clustering. Experimental results demonstrate that the proposed method outperforms existing mainstream multi-view clustering approaches on multiple benchmark datasets.

## Acknowledgments

This work was supported by the National Natural Science Foundation of China under Grants 62576067 and 62501226, the National Key Research and Development Program of China under Grant 2024YFB4710800, the Liaoning Provincial Submarine Environment Science Data Center Foundation under Grant 2025JH27/10100003, the Ministry of Industry and Information Technology of the People's Republic of China under Grant 18Q-25-06, the Liaoning Provincial Natural Science Foundation under Grants 2025-YQ-01 and 2024-MS-012, the Dalian Science and Technology Talent Innovation Support Plan under Grant 2024RY010, and the Natural Science Foundation of Hebei Province under Grant F2025201037.

## Impact Statement

This paper presents work whose goal is to advance the field of Machine Learning. There are many potential societal consequences of our work, none which we feel must be specifically highlighted here.

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
