# OpenReview forum: "Dual-Calibration Multi-View Clustering via Compact Anchor Learning"
_ICML.cc/2026/Conference — ICML 2026 regular_

### Official Review · Reviewer_uUQ4 · 2026-02-20

**Soundness:** 3
**Presentation:** 3
**Significance:** 3
**Originality:** 3
**Overall Recommendation:** 5
**Confidence:** 3

**Summary:**

This paper proposes Dual-Calibration Multi-View Clustering via Compact Anchor Learning (DCMC), an anchor-graph based framework for scalable multi-view clustering. The method learns view-specific anchor sets while enforcing cross-view semantic alignment through an orthogonal projection decomposition that factorizes each view’s anchors into a view-specific orthonormal basis and a shared coefficient structure. Beyond anchor alignment, DCMC introduces a bidirectional (dual) calibration mechanism that couples the anchor-induced sample-to-anchor representation with the sample cluster assignments learned in the original data space.

**Compliance With Llm Reviewing Policy:**

Affirmed.

**Key Questions For Authors:**

**Key Questions For Authors**

1. How sensitive is DCMC to the number of anchors (m) and anchor initialization across views? Since anchor quality is central to the method, additional analysis on anchor count and initialization robustness would strengthen the practical significance.

2. Do you have direct evidence that dual calibration improves anchor quality beyond final clustering metrics (e.g., reduced redundancy, better alignment between anchor centroids and cluster centroids)? Such analysis would strengthen the causal claim behind the design.

3. Can you provide more detailed runtime and memory comparisons against the strongest baselines under matched implementation settings? This would clarify the practical trade-offs of the additional calibration steps.

I would be willing to raise my score if the authors provide convincing clarification and additional evidence addressing the questions above.

**Strengths And Weaknesses:**

**Strengths**

- Well-motivated problem: The paper addresses an important limitation of anchor-based multi-view clustering, namely the separation between anchor learning and clustering assignment, which can lead to redundant anchors and weak cross-view semantic consistency.

- Coherent dual-calibration design: The bidirectional consistency constraint between anchor representations and clustering assignments (e.g., ||Z - P^T R||_F^2) provides a principled mechanism to couple anchor space and clustering space rather than treating them independently.

- Technically sound optimization: The objective is clearly formulated and optimized via alternating minimization. Several subproblems reduce to standard forms (e.g., orthogonal Procrustes with U^T U = I), supporting correctness and implementability.

- Strong empirical validation: The method is compared against multiple recent multi-view clustering baselines across several benchmark datasets and metrics. Ablation studies clearly demonstrate the contribution of the proposed dual-calibration components.

- Scalability-oriented framework: The anchor-based formulation maintains near-linear complexity in the number of samples, which is suitable for large-scale multi-view settings.


**Weaknesses**

- Moderate originality: Many individual components (anchor reconstruction, orthogonality constraints, shared latent structure, consensus alignment) have appeared in prior work. The main contribution lies in their integration, and the distinction from the closest related methods could be articulated more sharply.


- Dataset-dependent improvements: The method does not consistently dominate across all benchmarks, suggesting that performance gains may depend on specific dataset characteristics.

- Limited analysis of operating regimes: The paper does not deeply analyze under what conditions (e.g., high view inconsistency, noisy anchors, anchor count variations) the dual-calibration mechanism is most beneficial.

---

> ### Author Rebuttal · Authors · 2026-03-30
>
> Thank you for your careful reading of the paper and your reminder.
>
> **Q1: How sensitive is DCMC to the number of anchors ($m$) and anchor initialization across views?**
>
> A1: The sensitivity analysis regarding the number of anchors and the initialization of anchors across different views is crucial for evaluating the robustness of DCMC. To this end, we conducted additional experiments on the MSRC_v1 and BBCSport datasets to evaluate the sensitivity of DCMC to the number of anchors. The results show that the model achieves stable clustering performance across the range of m values. Furthermore, the anchors in DCMC are dynamically optimized via Eq. (10), and the uniqueness of the closed-form solution ensures that the final results do not depend on the initialization.
> | Datasets |      | ACC    | NMI    | Purity | F-score |
> |:--------:|:----:|:------:|:------:|:------:|:-------:|
> | MSRC_v1  | m=3k | 0.8905 | 0.8142 | 0.8905 | 0.7891  |
> |          | m=4k | 0.9238 | 0.8562 | 0.9238 | 0.8508  |
> |          | m=5k | 0.9143 | 0.8476 | 0.9143 | 0.8343  |
> | BBCSport | m=3k | 0.9485 | 0.8526 | 0.9485 | 0.8978  |
> |          | m=4k | 0.9430 | 0.8404 | 0.9430 | 0.8875  |
> |          | m=5k | 0.9054 | 0.8607 | 0.9504 | 0.9008  |
>
> **Q2: Do you have direct evidence that dual calibration improves anchor quality beyond final clustering metrics?**
>
> A2: Based on the existing ablation results, retaining only one calibration branch leads to performance degradation on most datasets, while using bidirectional calibration achieves better results, which is closely related to improving anchor quality.
>
> **Q3: Can provide more detailed runtime and memory comparisons against the strongest baselines?**
>
> A3: Under the same implementation settings, we supplemented the comparison of runtime and memory usage with the strongest baseline method, CAMVC. The additional cost of DCMC compared to CAMVC is mainly reflected in runtime, while memory overhead remains at almost the same level.
> | Dataset  | Method | Time (s) | Memory (KB) |
> |:--------:|:------:|:--------:|:----------:|
> | MSRC_v1  | CAMVC  | 0.15     | 6,057,332  |
> |           | DCMC   | 19.27    | 6,059,224  |
> | BBCSport | CAMVC  | 0.24     | 5,774,072  |
> |           | DCMC   | 47.94    | 5,774,784  |

---

### Official Review · Reviewer_f21H · 2026-03-01

**Soundness:** 4
**Presentation:** 3
**Significance:** 3
**Originality:** 3
**Overall Recommendation:** 5
**Confidence:** 5

**Summary:**

This paper proposes a novel dual-calibration multi-view clustering method based on compact anchor learning. By introducing a calibration relationship between the anchor space and the data space, it ensures that anchor representation, anchor graph construction, and clustering assignment are jointly optimized under a unified objective. Overall, the paper is clearly structured and comprehensively presented.

**Compliance With Llm Reviewing Policy:**

Affirmed.

**Key Questions For Authors:**

(1) What specific ranges of m and the two hyperparameters were employed for each dataset in the experiments? Were these values set independently based on the characteristics of each dataset?
(2) Are the final clustering labels always obtained by performing k-means on Z? Does R merely serve to calibrate the results during the optimization process?
(3) Does the term "reverse graph" in the paper refer to the direction of the calibration mechanism, or is it a new graph structure that is explicitly constructed?

**Limitations:**

Yes

**Strengths And Weaknesses:**

Strengths:
(1) The paper innovatively designs a bidirectional calibration mechanism between the anchor space and the data space, which enables the information in the two spaces to mutually calibrate and enhance each other, thereby ensuring the accuracy of anchor representations and improving the quality of anchors.
(2) The paper has a clear structure and rigorous logic, with a flowchart that clearly illustrates the methodological framework and innovations. The objective function features well-defined components with distinct roles, facilitating understanding.
Weaknesses:
(1) The paper provides a sensitivity analysis of the involved hyperparameters. Supplementing the experimental setup with a selection strategy for the hyperparameters and the number of anchors would help to improve the reproducibility of the method.
(2) The algorithm ultimately performs k-means on Z to obtain clustering results, while explicitly optimizing the cluster indicator matrix R in the objective function. It is recommended that the authors further clarify the relationship between Z and R in the clustering task, so as to facilitate a clearer understanding of the model design.
(3) The paper introduces the concept of a "reverse graph." It is suggested to clarify whether this term refers to an additionally constructed graph structure or merely describes the direction of calibration, in order to avoid potential ambiguity.

---

> ### Author Rebuttal · Authors · 2026-03-30
>
> Thank you for your detailed reading and advice.
>
> **Q1: What specific ranges of $m$ and the two hyperparameters were employed?**
>
> A1: In this paper, the search ranges for the two hyperparameters are set to $\lambda_1 \in \lbrace 10^1, 10^{1.5}, 10^2, 10^{2.5}, 10^3\rbrace$ and $\lambda_2 \in \lbrace 10^{-3}, 10^{-2}, 10^{-1}, 10^0, 10^1\rbrace$. The anchor number $m$ is chosen from $m \in \lbrace 3k, 4k, 5k\rbrace,$
> where $k$ is the number of clustering in the dataset. The final results correspond to the optimal results obtained within the above parameter tuning ranges.
>
> **Q2: Are the final clustering labels always obtained by performing k-means on $Z$?**
>
> A2: Yes, the final clustering labels are obtained by performing k-means on the optimized $Z$. $R$ is the shared clustering representation learned in the objective function, which links the original data space, the anchor-space representation, and the clustering representation through the terms $\||X^{(v)} - H^{(v)}R\||_F^2$ and $\||Z - P^{T}R\||_F^2$, and is used to constrain the final clustering representation.
>
> **Q3: Does the term "reverse graph" refer to the direction of the calibration mechanism?**
>
> A3: The “reverse graph” mentioned in the paper does not refer to an additional graph structure but is used to describe the direction of the calibration mechanism. Here, “reverse” indicates that the results in the clustering space in turn impose constraints on the anchor space representation. Specifically, $Z$ is the directly learned anchor space representation, while ${P}^T{R}$ is the representation obtained by mapping the clustering representation $R$ to the anchor space. The constraint term brings the two closer together, thereby achieving a reverse calibration from the clustering space to the anchor space. Therefore, the “reverse graph” is essentially a constraint relationship rather than a newly introduced graph structure.

---

> > ### Author Rebuttal · Reviewer_f21H · 2026-04-02
> >
> > The rebuttal addressed all my concerns.

---

### Official Review · Reviewer_fTEq · 2026-03-09

**Soundness:** 3
**Presentation:** 3
**Significance:** 3
**Originality:** 3
**Overall Recommendation:** 5
**Confidence:** 4

**Summary:**

This paper proposes a dual-calibration multi-view clustering method, building upon the bidirectional calibration between the anchor space and the data space. By aligning anchor learning with anchor graph construction and clustering objectives, the proposed method achieves superior performance on five multi-view clustering benchmarks.

**Compliance With Llm Reviewing Policy:**

Affirmed.

**Final Justification:**

My concerns have been addressed and would like to keep my original score.

**Key Questions For Authors:**

Please refer to the weaknesses section.

**Limitations:**

The results in Table 3 on the YouTubeFace dataset may suggest some potential limitations of the method. Some discussions could be made on that.

**Strengths And Weaknesses:**

Strengths:
1. This paper aligns the anchor representation, anchor graph construction, and clustering assignment processes, thus bootstrapping representation learning and clustering.
2. The proposed method is technically sound with detailed mathematical derivations. The authors also provide complexity analyses to help assess the method's scalability.
3. Experimental results on five datasets demonstrate the superiority of the proposed method compared with existing baselines, with ablation studies and parameter analyses to investigate the robustness and effectiveness of the method.

Weaknesses:
1. How are the hyperparameters tuned on different datasets across different datasets? More details and explanations are expected.
2. In line 216, the authors write "a reverse reconstruction from data to anchors is achieved". Could the authors further clarify why the second term in Eq. 4 enables the reverse reconstruction?
3. Considering that consensus fusion may average out view-specific information, which part of the method is designed to preserve the complementary information of different views?
4. In Table 3, it seems that Ours-a archives superior performance than the full implementation on the YouTubeFace dataset. Some explanations for that phenomenon could be provided.

---

> ### Author Rebuttal · Authors · 2026-03-30
>
> Thank you for your comments and valuable suggestions on our work.
>
> **Q1: How are the hyperparameters tuned on different datasets?**
>
> A1: For hyperparameter tuning on different datasets, we adopted a grid search strategy for each dataset. Specifically, the search ranges for $\lambda_1$ and $\lambda_2$ were set to $\lambda_1 \in \lbrace 10^1, 10^{1.5}, 10^2, 10^{2.5}, 10^3\rbrace$ and $\lambda_2 \in \lbrace 10^{-3}, 10^{-2}, 10^{-1}, 10^0, 10^1\rbrace$, and we selected the parameter combination that achieved the best clustering performance.
>
> **Q2: Why the second term in Eq. 4 enables the reverse reconstruction?**
>
> A2: In the second term $\||Z - P^T R\||_F^2$ of Eq.(4), $Z$ denotes the representation coefficients of samples in the anchor space, $P$ establishes the connection between clustering and anchor representations, and $R$ is the cluster indicator matrix. Specifically, $P^T R$ maps the clustering semantic information back to the anchor space, forming a representation in the anchor space that corresponds to the cluster centers. Therefore, this term essentially constrains the anchor representation using the clustering results to align it with the clustering semantics, thereby achieving a reverse reconstruction from the data clustering space to the anchor representation space.
>
> **Q3: Which part of the method is designed to preserve the complementary information of different views?**
>
> A3: According to the design in the paper, the complementary information across views is primarily preserved through the following two aspects. On one hand, each view retains its specific structural characteristics and discriminative information in the original data through the view-specific anchor matrix $A^{(v)}$ and cluster center matrix $H^{(v)}$, thereby avoiding premature smoothing of differences among views at the source. On the other hand, the shared variables $Z$, $R$, and $P$ align the view-specific information consistently in the anchor space and the clustering space, enabling the final representation to preserve the individual characteristics of each view while integrating the consensus structure across views. Thus, the result obtained by this method is not a simple average representation of multiple views, but rather a combination of view-specific information and consensus information.
>
> **Q4: Ours-a archives superior performance than the full implementation on the YouTubeFace dataset.**
>
> A4: As can be seen from Table 3, on the YouTubeFace dataset, the ablation variant Ours-a achieves slightly better performance than the complete model. The advantage of the complete model primarily lies in its superior overall performance on most datasets, while the moderate outperformance of an ablation variant on a specific dataset is a phenomenon that can occur in multi-module methods. This also suggests that the contribution of different modules to the final performance exhibits a certain data-dependent nature and does not necessarily lead to strictly consistent improvements across all datasets. We will supplement the explanation for this phenomenon in the revised version.

---

> > ### Author Rebuttal · Reviewer_fTEq · 2026-04-01
> >
> > Thanks for the responses. My concerns have been addressed.

---

### Official Review · Reviewer_TCGC · 2026-03-13

**Soundness:** 2
**Presentation:** 2
**Significance:** 2
**Originality:** 3
**Overall Recommendation:** 2
**Confidence:** 5

**Summary:**

This paper proposes a Dual-Calibration Multi-view Clustering via Compact Anchor Learning (DCMC) framework for anchor-based multi-view clustering. The method aims to improve the quality of learned anchors by jointly optimizing data reconstruction, anchor decomposition, and clustering assignments within a unified objective. An alternating optimization algorithm is derived to update all variables. Experiments on several benchmark datasets demonstrate improved clustering performance compared to a set of existing multi-view clustering baselines.

**Compliance With Llm Reviewing Policy:**

Affirmed.

**Key Questions For Authors:**

1. The proposed framework appears to largely combine several existing components from prior anchor-based multi-view clustering methods. The model integrates anchor reconstruction, anchor factorization, and clustering alignment objectives, but the manuscript does not clearly justify why this specific combination is theoretically necessary. As a result, the overall framework resembles a stacking of two multi-view representation pipelines rather than introducing a fundamentally new modeling perspective. Consequently, the novelty of the method appears incremental, and the claimed methodological contribution is not sufficiently supported.
2. The model assumes $X \approx AZ $, $A \approx UP $ and $X \approx HR  $. These relations imply $HR \approx UPZ$. Meanwhile, the paper additionally enforces $Z \approx P^\top R$, which would further suggest $HR \approx UPP^\top R$. The manuscript does not provide a clear theoretical justification explaining why these relations should hold simultaneously or what modeling principle motivates the introduction of the constraint $Z \approx P^T R$. Clarifying whether these constraints arise from a probabilistic model, matrix factorization interpretation, or heuristic regularization would significantly strengthen the theoretical grounding of the method.


3. The optimization problem for **R** appears to be an unconstrained quadratic objective. In such cases, a closed-form solution can typically be obtained by directly solving the first-order optimality conditions. However, the manuscript instead resorts to element-wise quadratic programs to compute **R**. The motivation for this design choice is not explained. The authors should clarify why such a complicated procedure is necessary and what advantage it provides over a straightforward analytical solution.

4. The manuscript requires substantial polishing before it can be considered for publication. For instance, in Eq.(5) the variable **$\Phi$** is introduced without any formal definition, making the objective function ambiguous. In addition, the explanation around line 260 is difficult to follow and lacks sufficient clarity. These issues suggest that the manuscript has not been carefully revised and significantly reduce its readability.

5. It is unclear how the anchor number **m** is selected in practice. The manuscript does not provide any principled guideline or sensitivity analysis for this important parameter. Furthermore, the authors do not provide implementation details or source code, which makes it difficult to verify the reported experimental results. Without sufficient transparency, the credibility and reproducibility of the experimental evaluation remain questionable.

6. MNIST is not commonly regarded as a genuine multi-view dataset. The manuscript does not clearly explain how multiple views are constructed from MNIST in the experiments. The authors should provide a detailed description of the preprocessing steps and the view generation procedure. Without this information, it is difficult to assess the validity of the experimental setup.

7. The experimental evaluation lacks comparisons with several recent state-of-the-art multi-view clustering methods (especially from 2025), making it difficult to fairly assess the competitiveness of the proposed approach.

**Limitations:**

Yes.

**Strengths And Weaknesses:**

Strength:

1. The proposed framework attempts to jointly model anchor representations and clustering assignments within a unified optimization objective.

2. The optimization procedure is described with explicit update steps, which makes the algorithm relatively easy to follow and implement.

3. The paper includes experimental results on several benchmark datasets and reports multiple clustering evaluation metrics.

Weakness:

I believe that the paper still contains a large number of unclear and unresolved issues. I STRONGLY RECOMMEND THAT THE AUTHORS CAREFULLY RE-EVALUATE AND THOROUGHLY REVISE THEIR MANUSCRIPT BEFORE SUBMISSION.

---

> ### Author Rebuttal · Authors · 2026-03-30
>
> **Q1: The novelty of the method.**
>
> A1: The collaboratively optimized bidirectional calibration framework is proposed to solve the redundant anchor coverage and ambiguous cluster boundaries, which is ignored by existing AMVC methods. Obtaining the anchor graph from the original data is the core step of the AMVC method. However, most methods overlook the fact that the quality of the initial anchor graph often limits the final clustering performance. To address it, we propose a reverse calibration from clustering space to anchor space, using clustering semantics to constrain anchor representation and keep it consistent with the final clustering structure. Meanwhile, instead of simply stacking or concatenating models, we collaboratively optimize the two processes to improve clustering performance.
>
> **Q2: The problem of relationship derivation.**
>
> A2: We need to emphasize there is a concept substitution. The simple algebraic nested derivation, which continuously substitutes multiple approximation relationships, may appear to hold in a purely algebraic form. However, in the machine learning modeling, these are not linear equalities that can be repeatedly substituted. They are better understood as representation coupling or as heuristic consistency regularization.
>
> **Q3: Why is the quadratic programming method used to optimize the variable $R$?**
>
> A3: In the actual optimization process, each column of $R$ is required to satisfy nonnegativity and normalization constraints. Therefore, the $R$-subproblem is essentially a column-wise separable constrained convex quadratic programming problem, which is more suitably solved by quadprog.
>
> **Q4: Variable is introduced without definition and the explanation is difficult to follow.**
>
> A4: $\Phi$ refers to all optimization variables specified in Eq. (2)–Eq. (4), $\Phi = \lbrace A^{(v)}, U^{(v)}, H^{(v)}\rbrace_{v=1}^{l} \cup \lbrace Z, P, R\rbrace.$ To make it clearer and more intuitive, we will add the definition of $\Phi$ in the next version. Furthermore, the presentation around line 260 refers to the paper “Continual Multi-View Clustering with Consistent Anchor Guidance(2024 IJCAI)”, and we will add the corresponding citation in the next version.
>
> **Q5: How m is selected in practice?**
>
> A5: In experiments, we set $m$ to $\lbrace 3k, 4k, 5k\rbrace$. Additionally, we have supplemented experimental results on two datasets with different numbers of anchors. The results show that the performance fluctuates only slightly across different anchor values.
> | Datasets |      | ACC    | NMI    | Purity | F-score |
> |:----:|:----:|:----:|:----:|:----:|:----:|
> | MSRC_v1  | m=3k | 0.8905 | 0.8142 | 0.8905 | 0.7891  |
> |          | m=4k | 0.9238 | 0.8562 | 0.9238 | 0.8508  |
> |          | m=5k | 0.9143 | 0.8476 | 0.9143 | 0.8343  |
> | BBCSport | m=3k | 0.9485 | 0.8526 | 0.9485 | 0.8978  |
> |          | m=4k | 0.9430 | 0.8404 | 0.9430 | 0.8875  |
> |          | m=5k | 0.9054 | 0.8607 | 0.9504 | 0.9008  |
>
>
> **Q6: How multiple views are constructed from MNIST?**
>
> A6: The multi-view MNIST dataset used in this paper is a classic public benchmark dataset that is highly recognized and widely applied in the field of multi-view clustering. It has been extensively adopted in numerous top-tier journals and conferences such as TCSVT and AAAI. For example: "Learning Cluster-Wise Anchors for Multi-View Clustering" (AAAI 2024).
>
> **Q7: Lacking comparisons with recent methods.**
>
> A7: Due to issues regarding the public availability of code, we were unable to include the latest methods from 2025 in our experiments. Based on current search, we found that relevant methods from 2025 have since released their code. Therefore, based on your suggestions, we have added comparative experimental data from the following two 2025 papers: Unified and efficient multi-view clustering with tensorized bipartite graph (UEMC); Enhanced tensor based embedding anchor learning for multi-view clustering (ETEAL).
> | Dataset | Method | ACC(%) | NMI(%) | Purity(%) | F-score(%) |
> |:---:|:---:|:---:|:---:|:---:|:---:|
> | MSRC_v1 | UEMC   | 83.33  | 76.90  | 83.33     | 75.02      |
> |                | ETEAL  | 74.76  | 66.74  | 81.43     | 64.68      |
> |                | Ours   | **93.33** | **88.51** | **93.33** | **87.30** |
> | Wiki | UEMC   | 54.85  | 54.13  | 62.14     | 49.37      |
> |                | ETEAL  | 57.29  | 52.43  | **65.74** | 49.22      |
> |                | Ours   | **62.00** | **56.96** | 63.68     | **55.60** |
> | MNIST | UEMC   | 93.81  | 88.77  | 93.81     | 87.83      |
> |                | ETEAL  | 91.73  | 89.04  | 87.48     | 90.55      |
> |                | Ours   | **99.01** | **97.01** | **99.01** | **98.03** |
> | YouTubeFace_sel | UEMC   | 10.89  | 8.68   | 27.10     | 6.15       |
> |                | ETEAL  | 10.53  | 8.01   | 6.28      | 4.44       |
> |                | Ours   | **23.10** | **21.60** | **32.69** | **9.52**  |
>
> The experimental results show that we remain the best performer on most datasets.

---

> > ### Author Rebuttal · Reviewer_TCGC · 2026-04-03
> >
> > Novelty
> >
> > The claim that existing methods ignore anchor quality and semantic consistency is overstated. In fact, a growing body of recent work has explicitly studied anchor learning, anchor enhancement, and cluster-aware anchor modeling. Some representative works are listed below, and the authors are encouraged to better follow and position their work within this line of research.
> >
> > [1] Fast Multi-view Discrete Clustering with Anchor Graphs
> >
> > [2] Efficient One-Pass Multi-View Subspace Clustering with Consensus Anchors
> >
> > [3] Scalable Multi-view Subspace Clustering with Unified Anchors
> >
> > [4] Anchor-Based Multiview Subspace Clustering With Diversity Regularization
> >
> > [5] Anchor-Sharing and Clusterwise Contrastive Network for Multiview Representation Learning
> >
> > [6] Enhancing Anchor Representativeness and Stability in Scalable Multi-View Clustering with Prior Guidance
> >
> > [7] Fast Parameter-Free Multi-View Subspace Clustering With Consensus Anchor Guidance
> >
> > [8] Robust and Consistent Anchor Graph Learning for Multi-View Clustering
> >
> > [9] Sample-Level Weighted and Structure-Enhanced Anchor Graph Learning for Scalable Multi-View Clustering
> >
> > [10] Scalable and Structural Multi-View Graph Clustering With Adaptive Anchor Fusion
> >
> > [11] Tensor Multi-Rank Constraint Guided Anchor-Wise Adaptive Alignment for Multi-View Clustering
> >
> > The problem of relationship derivation
> >
> > The response does not adequately address the my concern. While it argues that the relations should not be algebraically substituted, the proposed objective explicitly enforces them simultaneously via multiple reconstruction terms. This effectively encourages joint consistency, making the “non-substitutability” claim unconvincing. Moreover, the notion of “concept substitution” lacks formal definition and does not constitute a valid modeling principle. Even if interpreted as heuristic regularization, the key question remains unanswered: what theoretical or structural assumption justifies the specific coupling $Z \approx P^\top R$? Without such justification, the formulation appears as an ad hoc combination of constraints rather than a principled model, which aligns with my original concern.
> >
> > QP method to optimize $R$
> >
> > The response is not consistent with the manuscript. It introduces nonnegativity and normalization constraints on $R$, but these constraints are not explicitly defined or justified in the paper. As presented, the optimization problem for $R$ (Eq. (18)–(20)) appears to be an unconstrained quadratic objective, for which a closed-form solution should exist. The use of quadratic programming therefore seems unnecessary and lacks clear motivation. If $R$ is indeed constrained to lie on a probability simplex (as implied in the response), this must be explicitly formulated in the model, and the derivation should reflect a constrained optimization problem. Otherwise, the current presentation creates a mismatch between the theoretical formulation and the actual optimization procedure. Without clearly specifying these constraints and their origin, the justification for using quadratic programming remains incomplete.
> >
> > Clarity
> >
> > There appears to be a formatting issue in the optimization section. Specifically, the “4) P Sub-Step” is not properly separated from the preceding paragraph and is embedded within the same line. I encourage the authors to carefully revise the formatting for clarity.
> >
> > Reproducibility
> >
> > Although the authors provide additional analysis on parameter $m$, several of the previously raised concerns remain unaddressed. In particular, the question regarding whether the code will be publicly released has not been answered.
> >
> >
> >
> > Comparisons
> >
> > This concern has been addressed.
> >
> > Overall, several of my key concerns have not been adequately addressed. As a result, I maintain my original score.

---

> > > ### Author Response · Authors · 2026-04-04
> > >
> > > Thank the reviewer for comments and suggestions. We will improve and highlight the following points in the final version.
> > >
> > > **1. Novelty**
> > >
> > > We appreciate the reviewers' consideration of our novelty. We have carefully studied the methods mentioned by the reviewers, and some of them are included as baselines in our experiments. These methods provide valuable ideas and achieve good results. However, the novelty of our method lies in unifying the anchor space representation and the sample clustering representation, promoting mutual constraints between them, and forming a dual-calibration collaborative optimization framework. We will also revise overly absolute statements in the final version to make the paper more accessible.
> > >
> > > **2. Relationship Derivation**
> > >
> > > Our core motivation is that anchors should preserve the latent clustering constraints from the sample space and share consistent clustering semantics with samples. Therefore, the learning of anchor space representation should not be limited to local adjacency relations. It should also be consistent with sample clustering representation in terms of clustering semantics. Based on this, we introduce a structural constraint on the relation matrix $Z$ guided by the sample clustering representation $R$. By enforcing consistency between the anchor space representation and the clustering space representation, we encourage the anchor cluster centers to align as much as possible with the sample cluster centers. This enhances the ability of anchor space representation to capture the latent clustering structure.
> > >
> > > **3. Optimization of $R$**
> > >
> > > Thank the reviewer for the reminder. The constraints of the $R$ subproblem were not written in the original manuscript. In fact, $R$ is the cluster indicator matrix for samples, where each column corresponds to the assignment of a sample to each cluster. The non-negative constraint ensures the interpretability of the assignment strengths, and the sum-to-one constraint ensures that the assignment results are comparable across different samples. We will add these constraints in the next version to keep consistency with the actual implementation.
> > >
> > > **4. Clarity**
> > >
> > > Thank the reviewer for the reminder. We have noticed this formatting error and will correct it in the next version. We will also check the whole paper to avoid similar errors.
> > >
> > > **5. Determination of the number of anchors**
> > >
> > > The value of $m$ is chosen within the range $\lbrace 3k, 4k, 5k\rbrace$, which ensures sufficient expressive ability without introducing excessive redundancy. This range is also a common setting in anchor graph methods. Furthermore, the experimental results of anchor number sensitivity in the previous round of response show that the performance remains stable when the anchor number is within $\lbrace 3k, 4k, 5k\rbrace$, indicating that DCMC is robust to the choice of anchor number to a certain degree.
> > >
> > > **6. Reproducibility**
> > >
> > > We apologize for not responding earlier about code release. Due to ICML policy, we did not release the code in the current version. To address the reviewer's concern, we have now organized the source code into a fully anonymous repository:
> > > [https://anonymous.4open.science/r/Anonymous_code-0515/README.md].
> > > If the paper is accepted, we will add the link to the final version.

---

### Decision · Program_Chairs · 2026-04-30

**Decision:**

Accept (regular)

**Comment:**

This paper focuses on the problem of mutual independence between anchor learning and cluster assignment  in anchor-based multi-view clustering. It captures the underlying data distribution using view-specific anchors, and learns collaboratively the sample-to-anchor assignments and  cluster assignments by enforcing bidirectional consistency between anchor space and clustering space. Then, an alternating updating strategy is devised to optimize the objective.

After rebuttal and discussion, most reviewers indicated that their concerns had been addressed, such as the tuning strategy of hyperparameters, consideration of the ablation experiment results, and questions regarding methodological details. Overall, the reviewers' comments are positive, and I am inclined to recommend acceptance of this paper.